# A New Potential Dietary Approach to Supply Micronutrients to Physically Active People through Consumption of Biofortified Vegetables

**DOI:** 10.3390/nu14142971

**Published:** 2022-07-20

**Authors:** Sara Baldassano, Maria Rita Polizzi, Leo Sabatino, Rosalia Caldarella, Andrea Macaluso, Angelina Alongi, Gaetano Felice Caldara, Vincenzo Ferrantelli, Sonya Vasto

**Affiliations:** 1Department of Biological, Chemical and Pharmaceutical Sciences and Technologies, University of Palermo, 90128 Palermo, Italy; mariarita.polizzi@community.unipa.it; 2Dipartimento Scienze Agrarie, Alimentari e Forestali (SAAF), University of Palermo, Viale Delle Scienze, Ed. 5, 90128 Palermo, Italy; leo.sabatino@unipa.it; 3Department of Laboratory Medicine, “P. Giaccone” University Hospital, 90128 Palermo, Italy; liacaldarella@virgilio.it; 4Experimental Zooprophylactic Institute of Sicily, Via Gino Marinuzzi 3, 90129 Palermo, Italy; andrea.macaluso@izssicilia.it (A.M.); alongi.angela@gmail.com (A.A.); gaetanofelice.caldara@unipa.it (G.F.C.); vincenzo.ferrantelli@izssicilia.it (V.F.); 5Euro-Mediterranean Institutes of Science and Technology (IEMEST), 90139 Palermo, Italy

**Keywords:** dietary supplements, micronutrients, molybdenum, vegetarian diets, iron homeostasis

## Abstract

Micronutrients are required in many reactions involved in physical activity and exercise. Most physically active people do not meet the body’s needs in terms of micronutrients through diet. The novelty of the present manuscript is the use of an innovative dietary approach to supply micronutrients to physically active people through biofortified food. Therefore, the key point of this study was to verify whether supplementation with biofortified vegetables—and specifically molybdenum (Mo)-enriched lettuce—in healthy volunteers affects essential regulators of body homeostasis and, specifically, hematological parameters, iron and lipid metabolism, and hepatic function. Twenty-four healthy volunteers were allocated in a double-blinded manner to either a control group that consumed lettuce, or the intervention group, which consumed Mo-enriched lettuce, for 12 days. Blood samples were collected at baseline (T0) and after 12 days (T1). We found that supplementation with Mo-enriched lettuce did not affect hematological parameters, liver function, or lipid metabolism, but significantly improved iron homeostasis by increasing non-binding hemoglobin iron by about 37% and transferrin saturation by about 42%, while proteins of iron metabolism (e.g., transferrin, ferritin, ceruloplasmin) were not affected. The serum molybdenum concentration increased by about 42%. In conclusion, this study shows that consumption of Mo-biofortified lettuce ameliorates iron homeostasis in healthy subjects, and suggests that it could be used as a new nutritional supplementation strategy to avoid iron deficiency in physically active people.

## 1. Introduction

Micronutrients play key roles in energy metabolism; therefore, their deficiency states may have profound consequences, especially for athletes [1]. In fact, a balanced diet, consumed in adequate quantities to meet energy requirements, supplies the required amounts of dietary minerals. However, most of the population—especially athletes—do not meet the body’s needs in terms of macronutrients and micronutrients to function satisfactorily through diet alone [2,3,4,5]. This is due not only to inadequate dietary intake (not all athletes get an appropriate energy intake from diet, and they may not eat a varied diet), but also to the higher physiological demand in the different stages of life, e.g., during childhood development and aging [6]. Furthermore, functional foods, due to the vegetable matrix, might increase the absorption and bioavailability of nutrients.

Molybdenum (Mo) is one of the nutritional micronutrients essential for the human body. Mo is a cofactor for four different enzymes: sulfite oxidase, xanthine oxidase, aldehyde oxidase, and the mitochondrial amidoxime-reducing component [7]. In each of them, Mo is bound to a very complex and organic component called molybdopterin, forming the entity molybdenum cofactor. Foods rich in Mo include nuts, legumes, and grains. However, Mo content in food is influenced by the soil. Therefore, for example, the content in meat depends on the forage of the animals, while that in plants depends on the regional richness of the soil [8]. The recommended dietary allowance of Mo is 45 μg/day [9]. The tolerable upper intake level is 2 mg/day. The biomonitoring equivalent (BE) of molybdenum in urine (considering the reference dose and tolerable daily intake exposure guidance values) is 22 μg/L, while the BE value associated with toxicity is 200–7500 μg/L [10].

Biofortification is considered a good strategy to deliver micronutrients to the populations and, among other agronomic practices [11,12,13,14,15], it represents a key technique to improve crop quality [16,17,18,19,20,21,22]. It differs from fortification because there is no addition of nutrient supplements to the foods during food processing; rather, plants are grown under specific conditions to be naturally more nutritive. Hence, the crops increase their nutritional value [23]—for example, in terms of micronutrients. Moreover, consumption of biofortified vegetables in individuals improves body health [24,25].

Iron is an essential dietary mineral because it is a key component of the hemoglobin protein, and is a cofactor of many enzymes involved in essential cell functions [26]; it is used by the body in many fundamental processes at the cellular level, such as energy production and oxygen transport. Therefore, iron homoeostasis is important in the general population to avoid iron deficiency, and is essential in athlete populations, where iron deficiency is reported in 15–35% of female and 3–11% of male athletes, and can compromise athletic performance [27].

Previous studies have shown that sustained-release preparations of molybdenized ferrous sulfate given as a supplement during pregnancy improve iron deficiency [28,29,30]. Therefore, we thought that biofortification of vegetables with molybdenum, if it impacts iron homeostasis, could offer an excellent opportunity to increase iron availability in the human body. It could reduce the negative effects of iron integration treatments—such as constipation and gastric effects—due to its vegetal-rich matrix. Therefore, the aim of the present study was to verify whether supplementation for 12 days with Mo-fortified vegetables—and specifically Mo-biofortified lettuce—influences iron homeostasis, hematic and lipid profiles, and liver function in a population of healthy individuals.

## 2. Materials and Methods

### 2.1. Experimental Design

The study employed a randomized, placebo-controlled design over a 12-day period, and is part of a larger program—the Nutri-Mo-Food project. The study was registered at Clinicaltrials.gov as NCT04985240. The protocol was approved by the Ethics Committee of Palermo University Hospital (Project Number 2/2020 AIFA CE 150109), and was conducted in accordance with the Declaration of Helsinki.

### 2.2. Participants

An a priori power calculation based on previous studies of hematological parameters [24,25,31], utilizing a level of statistical significance (α) of 5% and probability (β) of 20%, was used to estimate a sample size of eight participants. Twelve subjects were included in the study to reduce the risk of type-two errors and to enhance the evaluation power for secondary outcomes. As previously described, 24 healthy volunteers—12 males and 12 females—participated in the study, and provided written informed consent prior to study inclusion. Participant characteristics are shown in Table 1, while a flowchart of the recruitment, selection, and assignment of participants to groups is presented in Figure 1.

Participants were required to satisfactorily answer a health screening questionnaire, a daily food diary, and lifestyle information, and to be prepared to comply with study requirements. Participants with a known history of blood-related dysfunction; cardiac, gastrointestinal, or metabolic disorders; or recent viral infection were not eligible for the study. Additionally, anybody reporting the use of supplementation (e.g., molybdenum) or medication (including exogenous hormones for females) for >6 weeks before the start of the study, as well as those with known adverse or allergic reactions to dietary intake of lettuce, were excluded from the study. Pregnant and breastfeeding patients were not eligible for the study. The eligibility criteria of the study are summarized in Table 2. Participants were instructed by nutritionists and physicians to maintain the same dietary habits and lifestyle (including physical activity levels) throughout the intervention period. They were provided with example diaries and individually instructed in diary completion. As previously reported [24], food diaries were completed by participants in the first 8 days before starting the study and until the end of the study, as a means to monitor adherence and compliance, along with food habits and lifestyle changes.

### 2.3. Procedures of the Experimental Design

The procedures took place in the Nutrition, Age, and Bone (NABbio) laboratory of the STEBICEF Department at the University of Palermo, under controlled conditions. As previously reported [24], participants attended the laboratory, having had their last meal ~12 h before the appointment. Upon arrival, each participant’s anthropometric measurements (i.e., height, body mass, body fat) were recorded [32,33], and a venous whole-blood sample was collected in VACUTAINER and EDTA-K3 tubes between 7:00 and 8:00 a.m. [34,35]. At baseline, participants were allocated in a double-blinded manner to one of the two intervention groups. Each participant received three entire crops of lettuce, corresponding to about two kilograms of canasta lettuce, in order to consume 100 g each day, for a total of 12 days. For the storage conditions, it was suggested to wash and dry the lettuce and to store it in a domestic refrigerator (at 4 °C, in the drawers for storing vegetables) wrapped in a kitchen cotton towel. Before the consumption of the lettuce, the appearance of the leaves was examined—the color, the texture, and the smell. Lettuce with leaves of a bright green color, firm and hard, smooth, and with a good smell (decisive, aromatic) was considered good for consumption. If the leaves were yellow brown or gray and the smell was pungent (like stagnant water), the lettuce was considered not good for consumption. Six male and six female participants, for a total of twelve volunteers, were allocated to the control group and received control canasta lettuce, while six male and six female participants, for a total of twelve volunteers, were allocated to the molybdenum-biofortified group, and received canasta lettuce with molybdenum. As previously reported [20], lettuce enrichment with Mo was achieved through foliar spraying of molybdenum as sodium molybdate (Na_2_MoO_4_) during the period of growth. Specifically, the lettuce was supplied with a dose of 3 μmol Mo L^−1^ in the form of sodium molybdate. Therefore, as previously reported [24], the intervention group received 8 mg of Mo in 100 g of fresh lettuce, while the control group received 0.21 mg of Mo in 100 g of fresh lettuce, every day, for a total of 12 days. Blood sample collection was performed at baseline = T0 and at T1, after 12 days (Figure 2).

### 2.4. Analysis of Blood Samples

Blood sample analysis was performed at the central analysis laboratory of the Policlinico Hospital of the University of Palermo at T0 and T1. Whole blood samples were collected in tubes without any anticoagulant and fractionated by centrifugation at 1.300× *g* for 15 min at room temperature to obtain serum. The serum was used to measure the following chemical markers, using automated procedures according to standard, commercially available assays supplied by Roche Diagnostics, performed on the Roche COBAS c503: Iron, ferritin, transferrin saturation, and albumin (ALB) were measured by immunoturbidimetric assay, while ceruloplasmin and transferrin were measured by turbidimetric assay. Triglycerides (TG), total cholesterol, cholesterol LDL, alkaline phosphatase (AF), gamma-glutamyl transferase (GGT), total protein (TP), and HDL cholesterol were measured by colorimetric assay. Aspartate aminotransferase (AST) and alanine aminotransferase (ALT) were measured by the IFCC (International Federation of Clinical Chemistry and Laboratory Medicine) pyridoxal 5′-phosphate activation assay.

For determining blood cell counts and other blood parameters, the blood was collected in EDTA tubes and immediately inverted 10 times to mix the anticoagulant additive with the blood. The tubes were immediately processed using the Sysmex automated hematology analyzer (Roche Diagnostics, Milan, Italy), and the following parameters were determined: red blood cell count (RBC), white blood cell count (WBC), platelet count, hemoglobin, hematocrit, mean cell volume (MCV), mean corpuscular hemoglobin concentration (MCHC), mean corpuscular hemoglobin (MCH), red blood cell distribution width (RDW), and platelet count (PLT).

### 2.5. Analysis of Mo Concentration in Serum

The determination of Mo in samples of serum was performed by inductively coupled plasma mass spectrometry (7700x series ICP-MS, Agilent Technologies, Santa Monica, CA, USA). The extraction procedure and the ICP-MS analysis were carried out as described by Cammilleri et al. [36]. Briefly, 1 mL of the serum samples was digested using an Ultrawave digestion system (Milestone, Sorisole, Italy) with 3 mL of 60% (*V*/*V*) ultrapure nitric acid and 5 mL of deionized water in previously decontaminated polytetrafluoroethylene vessels. Digestion was conducted in an Ultrawave digester (Milestone, Sorisole, Italy). The samples were digested and diluted to 50 mL with ultrapure (Milli-Q) deionized water until the ICP-MS analysis. The instrument parameters were as follows: nebulizer carrier gas flow, 1.2 L/min; plasma gas flow, 15 L/min 15; reflected power, <5; RF power, 1550 W. For calibration, a certified reference standard from Agilent (USA) was used. The analysis was carried out on the basis of calibration curves constructed by the linear interpolation of at least 7 points corresponding to the readings of 7 standard solutions and white calibration, admitting a maximum error of 5% in the reading of the single standards, and a correlation coefficient r2 > 0.999. The validation parameters considered were LOD, LOQ, repeatability, recovery, and uncertainty. The LOD value of 0.0004 mg kg^−1^ and LOQ value of 0.0013 mg kg^−1^ were calculated on the basis of the results for 10 replicates of a serum sample spiked at 0.005 mg kg^−1^ (level 1), 10 replicates of a serum sample spiked at 0.010 mg kg^−1^ (level 2), and 10 replicates spiked at 0.05 mg kg^−1^ (level 3). Repeatability was 0.001 mg kg^−1^ at the first level, 0.001 mg kg^−1^ at the second level, and 0.004 mg kg^−1^ at the third level. Recovery of 104.3% was obtained using 10 replicates at each level. Uncertainty was ±0.001 mg kg^−1^ at the first level, ±0.001 mg kg^−1^ at the second level, and ±0.005 mg kg^−1^ at the third level. The measuring field was between 0.002 and 0.05 mg kg^−1^. 

### 2.6. Statistical Analyses

To compare the characteristics of the groups at baseline, we used Student’s *t*-test, while one-way ANOVA followed by Tukey’s post hoc test was used to compare differences between T0 and T1, using GraphPad Prism software. A *p*-value ≤ 0.05 was considered statistically significant for all tests. Data are presented as means ± standard deviations.

## 3. Results

The study enrolled a total of 24 individuals. The characteristics of the control and treated populations are listed in Table 1. The population enrolled was homogeneous, with no significant differences between the groups in terms of age, weight, height, lean or fat body mass, or visceral fat. They joined and completed the short-term intervention study in good health, with excellent compliance, and without drop-out.

### 3.1. Consumption of Mo-Biofortified Lettuce and Hematological Parameters

The hematological profile of the intervention group was compared with the control group at T0 (baseline), and after 12 days of lettuce consumption. The results of the comparison are shown in Table 3. No differences in the quality and quantity of red and white blood cell or platelet counts, hemoglobin, hematocrit, mean cell volume, corpuscular hemoglobin concentration, mean corpuscular hemoglobin, or red blood cell distribution width were observed between the control group and the Mo-treated group following the 12 days of intervention.

### 3.2. Consumption of Mo-Biofortified Lettuce and Iron Homeostasis

Different parameters involving iron metabolism were compared between the two study groups. Within the intervention group (Figure 3A,B, in purple), we observed an increase in non-hemoglobin blood iron concentration and transferrin saturation after 12 days of lettuce consumption with respect to the control group (Figure 3A,B, in green). The values of non-hemoglobin blood iron concentration and transferrin saturation were within the physiological range. The non-binding hemoglobin iron increased by about 37% with respect to the control group, while transferrin saturation increased by about 42% with respect to the control group. On the other hand, ferritin, transferrin, and ceruloplasmin concentrations were not influenced by the nutritional intervention. In the control group, no differences were reported in non-hemoglobin blood iron, transferrin saturation, ferritin, transferrin, or ceruloplasmin (Figure 3, in green).

### 3.3. Consumption of Mo-Biofortified Lettuce: Liver Function and Lipid Homeostasis

Liver function parameters were compared between the two groups (control and intervention). No differences were reported in aspartate aminotransferase, alanine aminotransferase, alkaline phosphatase, gamma-glutamyl transferase, albumin, or total protein in the intervention group following 12 days of short-term nutritional intervention (Table 4).

We then compared the lipid profiles of the two groups (control vs. intervention). No differences were observed between the control group and the Mo-treated group at baseline or following 12 days of short-term intervention in terms of triglycerides, total cholesterol, LDL cholesterol, or HDL cholesterol (Table 5).

### 3.4. Serum Mo Concentration

Table 6 shows the serum concentrations of Mo measured in the intervention group and the control group at T0 (baseline), and following 12 days of lettuce consumption. Serum concentrations of Mo were significantly increased in the intervention group at T1 with respect to the control group. Within the intervention group, we also observed a significant increase in Mo serum concentrations after 12 days of lettuce consumption with respect to T0. The serum molybdenum concentrations increased by about 42% in the intervention group compared to the control group.

## 4. Discussion

Primary prevention aims to prevent disease before it ever occurs. The present nutritional intervention, in a population of healthy individuals, evaluated the effects of supplementation with Mo-fortified lettuce for 12 days on iron homeostasis, hematic and lipid profiles, and liver function. These represent essential functions for the proper functioning of the human body, and are influenced by nutrition. This study aimed to investigate whether supplementation with Mo-fortified lettuce could be used as a new dietary approach to supply micronutrients to individuals to improve their health status, since no studies have investigated this to date.

We selected a healthy population by using specific selection criteria to exclude participants with disorders such as cardiac, gastrointestinal, or metabolic disorders, blood-related dysfunctions, and recent viral infections. We measured the hematic parameters in the control and treated groups before and at the end of the intervention in order to reduce the intraindividual and interindividual variability.

The idea to examine the effects of supplementation with Mo-fortified lettuce on iron homeostasis in a healthy population came from our recent study [24], where it was found that eating Mo-enriched lettuce for 12 days could positively impact glucose homeostasis. In particular, we observed an improvement in insulin sensitivity, as measured by HOMA2-%S, of fasting glucose, insulin, and insulin resistance index. Therefore, because glucose and systemic iron metabolism are strictly connected (i.e., adequate iron availability is essential for functional beta cells to regulate glucose homeostasis [37]), we decided to investigate whether the nutritional intervention with Mo-biofortified lettuce would impact iron metabolism.

It was found that Mo-enriched lettuce consumption affects systemic iron metabolism. In fact, 100 g of Mo-enriched lettuce, eaten every day, significantly increased non-binding hemoglobin iron concentration and transferrin saturation, within the physiological range. The results were confirmed by the lack of changes in blood iron concentration and transferrin saturation in the control group. Mo is a cofactor of the enzyme xanthine oxidase, which participates in the release of ferritin iron [38]. Thus, it is possible that the greater availability of Mo, due to the daily consumption of Mo-enriched lettuce, enhances the activity of the xanthine oxidase enzyme, and causes an increase in hematic iron. Therefore, the short-term nutritional intervention might influence iron homeostasis by increasing iron’s availability from the intestine in a physiological way, and then distributing iron to areas that need iron to properly function, such as the pancreas [37].

Thus, consumption of Mo-enriched lettuce can affect iron homeostasis and, in turn, this might improve glucose homeostasis, as we observed in our previous study [24]. In fact, the homeostasis of iron lies in the regulation of dietary iron absorption, because the body is unable to excrete iron [39], but this also tends to limit the absorption of iron. The presence of ferritin in the mucosal cells inhibits further absorption of iron from the intestine until the ferritin releases its iron into the bloodstream and again becomes apoferritin. Xanthine oxidase, present in the small intestine, acts during iron absorption by reducing iron of ferritin to a ferrous state and, thus, releases iron into the circulation [38]. After the exit as ferrous iron, it is converted to ferric iron and binds to transferrin before being distributed throughout the body. Our results are consistent with previous studies that reported the ability of Mo to enhance iron absorption from the intestine in women who were supplemented with a preparation of molybdenum ferrous sulfate to treat iron deficiency during pregnancy [28,29,30]. Supplementation with molybdenum has also been used for the treatment of anemia in the general population [40]. However, in our study, we did not find differences in hematocrit, hemoglobin level, or blood cell counts and volumes. This is probably due, on the one hand, to the fact that our volunteers were healthy and not suffering from iron deficiency. On the other hand, it might be possible that 12 days are not enough to show any difference in hematocrit parameters.

The quantity of Mo that was provided in our nutritional intervention was in line with the biomonitoring equivalent (BE) values recommended to protect against nutritional toxicity and deficits [10]. In fact, the Mo serum concentration that we measured was about 5 μg/L in the controls, and increased to 7 μg/L in the intervention group. The values were consistent with those reported in a recent study, which measured Mo serum concentrations by inductively coupled plasma mass spectrometry in a total population of 120 subjects (63 of whom were controls) [41], and found results far from 31 μg/L, which is the BE toxicity value reported for plasma and serum [10].

Mo has been reported to improve metabolic syndrome by acting on all of its components, including hypertriglyceridemia and HDL [42]. We did not observe any differences in triglycerides, total cholesterol, LDL cholesterol, or HDL cholesterol in the intervention group compared with the control group. Although it is possible to hypothesize that lipid profiles could be improved by prolonging the time of the nutritional intervention, it must be considered that our study enrolled a healthy population.

The liver plays a central role in human bodily metabolism, with hepatocytes carrying out most of the function [43]. Therefore, we also investigated the effects of the short-term nutritional intervention on liver function via the main liver functionality test, and specifically aspartate aminotransferase, alanine aminotransferase, alkaline phosphatase, gamma-glutamyl transferase, albumin, and total protein concentrations. No differences were observed between the two study groups, suggesting that consumption of Mo-enriched lettuce did not impact liver function in healthy subjects.

Ceruloplasmin and transferrin are synthesized by hepatocytes [37]. Ceruloplasmin facilitates cell iron egress, allowing the association of iron with transferrin, and the transport of iron toward the cells of the organism. Therefore, we then investigated whether consumption of Mo-enriched lettuce might impact these two proteins involved in iron homeostasis. The lack of changes in ceruloplasmin and transferrin concentrations suggests that hepatic synthesis of proteins involved in iron metabolism was not affected.

## 5. Conclusions

In the present study, we verified whether supplementation with biofortified vegetables—specifically, molybdenum (Mo)-enriched lettuce—in healthy volunteers would positively impact the essential regulators of bodily homeostasis and, specifically, hematological parameters, iron and lipid metabolism, and hepatic function. The purpose was to find an easy, natural, and innovative approach to supply micronutrients to physically active people via the diet, in an attempt to prevent micronutrient deficiency. The nutritional intervention, conducted in healthy volunteers by supplying 100 g of Mo-enriched lettuce for 12 days, suggested that biofortification of vegetables with molybdenum could offer an excellent opportunity to increase iron availability in the human body. Therefore, we believe that this kind of natural dietary supplementation could be of interest in the general population, but even more so in a physically active population, because systemic iron homeostasis is fundamentally important for athletes, with implications for sports performance. Iron is a component of various muscle cell enzymes, as well as of myoglobin, hemoglobin, and cytochrome. It plays a vital role in energy metabolism, transporting oxygen, and supporting the immune system. Insufficient reserves of iron in the body can reduce athletic performance, which may manifest as fatigue, exercise intolerance, or even cognitive function impairment. Supplementation with Mo-enriched lettuce increased serum molybdenum concentrations by about 42%, and significantly improved iron homeostasis by enhancing non-binding hemoglobin and transferrin saturation. This, in our opinion, is of particular interest for athletes, who could use this innovative nutritional approach to prevent the disturbances of systemic iron homeostasis that athletes experience—for example, during intense training periods characterized by higher training loads. Moreover, we think that this innovative nutritional approach could be used to reduce the risk of depletion of iron stores in specific athlete populations, such as women and adolescents, in whom the depletion of iron is especially high. The iron levels of physically active women, in some cases, are even twice as low as in non-trained humans. Our nutritional approach could prevent the occurrence of severe iron deficiency by reducing the negative effects of classic iron integration treatments—such as constipation and gastric effects—thanks to its vegetal-rich matrix. In conclusion, our study suggests that biofortification of vegetables with molybdenum could offer an excellent opportunity to avoid iron deficiency in physically active people by improving iron homeostasis.

## Figures and Tables

**Figure 1 nutrients-14-02971-f001:**
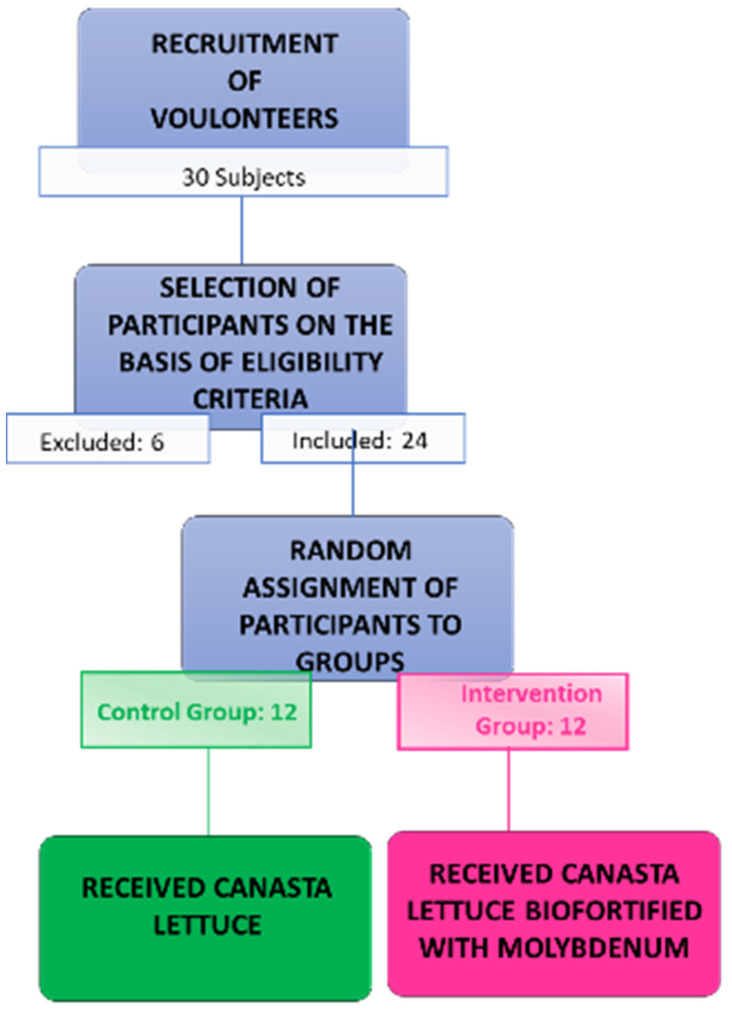
Flowchart of the recruitment process during the study period.

**Figure 2 nutrients-14-02971-f002:**
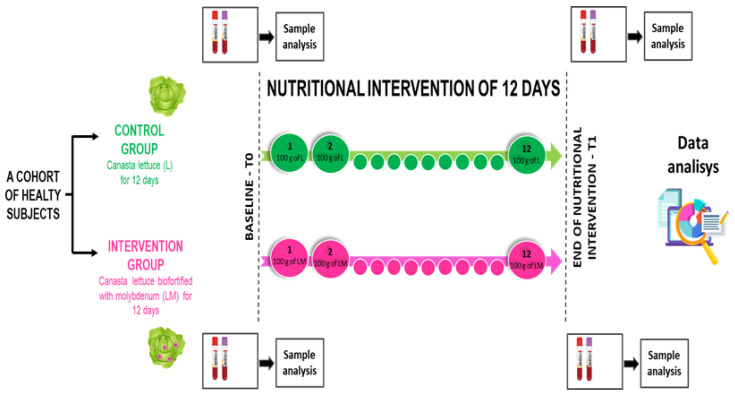
Flowchart of the nutritional interventional study during the period of 12 days of administration of canasta lettuce, with and without molybdenum biofortification. Green represents the control group, who ate canasta lettuce with no biofortification. Purple represents the treated group, who ate lettuce biofortified with molybdenum. Blood samples were collected at baseline (T0) and at the end of the nutritional intervention (T1).

**Figure 3 nutrients-14-02971-f003:**
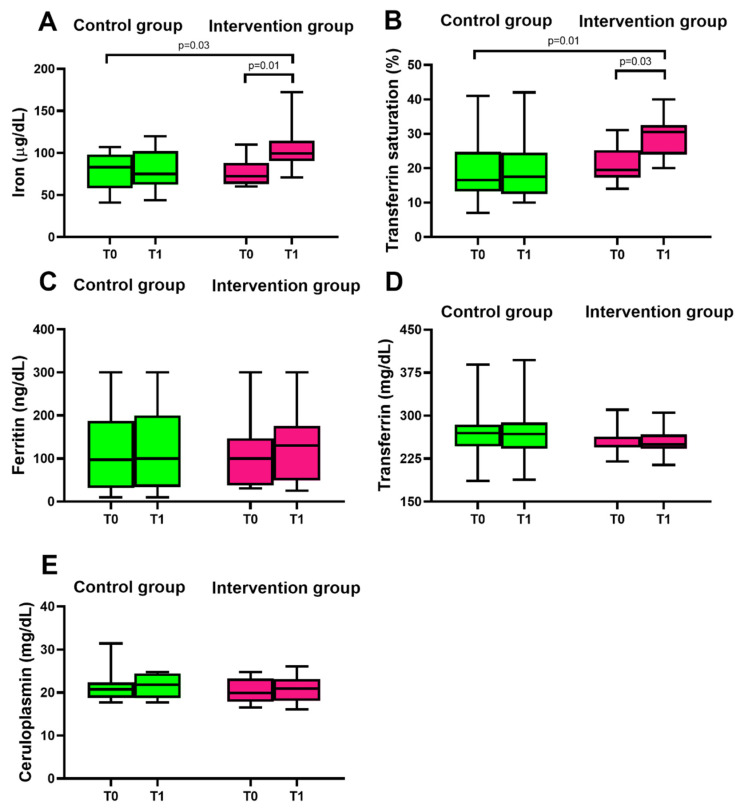
Iron metabolism in response to the 12-day nutritional intervention with control lettuce (green, control group) or Mo-biofortified lettuce (purple, intervention group). Panels show box-and-whisker plots: (**A**) iron, (**B**) transferrin saturation, (**C**) ferritin, (**D**) transferrin, and (**E**) ceruloplasmin concentrations taken at baseline (T0) and after 12 days (T1) in the two study groups; *p* < 0.05 was considered statistically significant. Iron concentration was significantly different in the control group (*p* = 0.03) with respect to the intervention group at T1, and within the intervention group at T0 (*p* = 0.01) with respect to T1. Transferrin saturation was significantly different in the control group (*p* = 0.01) with respect to the intervention group at T1, and within the intervention group at T0 (*p* = 0.03) with respect to T1.

**Table 1 nutrients-14-02971-t001:** Characteristics of participants at baseline; *n*: number of members in the group. Values are indicated as means ± standard deviations (SD); *p*-values higher than 0.05 mean that the change is not statistically significant.

Parameters	Control Group Age Range: 23–57 (*n* = 12), Mean ± SD	Intervention Group Age Range: 27–53 (*n* = 12), Mean ± SD	*p*-Value
Age	37.5 ± 13.5	40 ± 10	>0.05
Height(m)	1.68 ± 10	1.72 ± 9	>0.05
Weight (Kg)	69 ± 9.8	72 ± 13	>0.05
BMI (kg/m^2^)	24.3 ± 2.5	24.2 ± 2.8	>0.05
Visceral fat (%)	6.5 ± 3.3	7.2 ± 3.2	>0.05
Fat mass (%)	28.6 ± 7.3	26.5 ± 7.1	>0.05
Lean mass (%)	32.8 ± 5.8	33.2 ± 5.4	>0.05
Median age	38.5	39	>0.05

**Table 2 nutrients-14-02971-t002:** Eligibility criteria summary of the study.

Selection Criteria	Inclusion Criteria	Exclusion Criteria
Absence of blood-related dysfunction; cardiac, gastrointestinal, and metabolic disorders; recent viral infection; and food allergies	Volunteers of Italian ethnicity	Presence of chronic disease
Not taking medications	Age range: 18–60 years	Use of medication
Absence of obesity	Clinically healthy	Use of dietary supplements
Not taking supplements	Body mass index: 18.5–28 kg/m^2^	Breastfeeding, Pregnancy

**Table 3 nutrients-14-02971-t003:** Hematological profiles in response to the 12-day nutritional intervention with control or Mo-biofortified lettuce. Values are indicated as means ± standard deviations; *p*-values higher than 0.05 mean that the change is not statistically significant.

Parameters	Control Group			Intervention Group		
	Mean ± SD		*p*-Value	Mean ± SD		*p*-Value
	T0	T1		T0	T1	
WBC (10^3^/µL)	7.3 ± 2.0	7.0 ± 2.2	>0.05	7.1 ± 1.3	7.5 ± 1.8	>0.05
NEUT. (%)	57 ± 6.1	54.3 ± 7.2	>0.05	56.7 ± 6.5	58.7 ± 9.7	>0.05
LYMP. (%)	30.7 ± 3.7	34.1 ± 5.1	>0.05	31.7 ± 7.1	30.3 ± 9.6	>0.05
MON. (%)	8.3 ± 1.8	7.9 ± 1.5	>0.05	7.9 ± 0.9	7.7 ± 1.1	>0.05
EOS. (%)	2.2 ± 1.4	2.0 ± 1.0	>0.05	2.8 ± 1.9	2.3 ± 1.5	>0.05
BAS. (%)	0.7 ± 0.3	0.7 ± 0.4	>0.05	0.6 ± 0.3	0.7 ± 0.5	>0.05
NEUT. (10^3^/µL)	3.5 ± 1.3	3.6 ± 1.4	>0.05	3.2 ± 1.0	3.1 ± 1.0	>0.05
LYM. (10^3^/µL)	2.3 ± 0.8	2.4 ± 1.0	>0.05	2.2 ± 0.8	2.2 ± 0.8	>0.05
MON. (10^3^/µL)	0.6 ± 0.2	0.5 ± 0.1	>0.05	0.5 ± 0.2	0.6 ± 0.1	>0.05
EOS. (10^3^/µL)	0.2 ± 0.2	0.2 ± 0.1	>0.05	0.2 ± 0.1	0.2 ± 0.1	>0.05
BAS. (10^3^/µL)	0.0 ± 0.0	0.0 ± 0.0	>0.05	0.0 ± 0.0	0.0 ± 0.0	>0.05
RBC (10^6^/µL)	4.8 ± 0.6	4.8 ± 0.6	>0.05	4.8 ± 1.0	4.9 ± 0.6	>0.05
HGB (g/dL)	13.9 ± 1.1	13.2 ± 1.4	>0.05	14.4 ± 1.8	14.0 ± 1.6	>0.05
HCT %	39.1 ± 5.3	38.7 ± 4.6	>0.05	41.4 ± 6.4	40.5 ± 4.5	>0.05
MCV (fL)	83.2 ± 6.4	81.7 ± 6.7	>0.05	84.5 ± 7.3	82.9 ± 7.0	>0.05
MCH (pg)	27.6 ± 2.7	27.8 ± 2.7	>0.05	28.6 ± 3.2	28.7± 2.9	>0.05
MCHC (g/dL)	33.1 ± 1.4	33.6 ± 1.9	>0.05	33.2 ± 1.8	34.1 ± 1.7	>0.05
RDW (%)	13.5 ± 1.3	13.6 ± 1.3	>0.05	13.5 ± 1.3	13.5 ± 1.3	>0.05
RDW (fL)	40.0 ± 2.4	39.8 ± 2.8	>0.05	40.6 ± 2.9	40.2 ± 2.8	>0.05
PLT (10^3^/µL)	245.2 ± 42.8	232.7 ± 43.3	>0.05	262.0 ± 72.0	273.2 ± 47.4	>0.05

Acronyms: HCT, hematocrit; HGB, hemoglobin; MCH, mean corpuscular hemoglobin; MCHC, mean corpuscular hemoglobin concentration; MCV, mean cell volume; RBC: red blood cells; RDW, red blood cell distribution width; PLT, platelet; WBC, white blood cells.

**Table 4 nutrients-14-02971-t004:** Liver function in response to the 12-day nutritional intervention with control or Mo-biofortified lettuce. Values are indicated as means ± standard deviations; *p*-values higher than 0.05 mean that the change is not statistically significant, and indicate strong evidence for the null hypothesis.

Parameters	Control Group			Intervention Group		
	Mean ± SD		*p*-Value	Mean ± SD		*p*-Value
	T0	T1		T0	T1	
AST (U/L)	22.0 ± 4.5	19.0 ± 5.0	>0.05	20.7 ± 4.4	18.6 ± 6.9	>0.05
ALT (U/L)	19.9 ± 11.3	20.6 ± 9.1	>0.05	19.3 ± 8.5	12.7 ± 4.9	>0.05
ALP (U/L)	63.0 ± 10.0	58.5 ± 10.9	>0.05	65.5 ± 11.6	59.4 ± 8.8	>0.05
GGT (U/L)	15.2 ± 6.2	14.7 ± 5.1	>0.05	15.7 ± 6.4	15.7 ± 6.4	>0.05
TP (g/L)	71.1 ± 3.0	70.8 ± 3.9	>0.05	71.4 ± 2.9	70.7 ± 3.7	>0.05
ALB (g/L)	45.5 ± 3.5	46.5 ± 2.7	>0.05	44.1 ± 2.3	46.7 ± 2.7	>0.05

Acronyms: AST, aspartate aminotransferase; ALT, alanine transaminase; ALP, alkaline phosphatase; GGT, gamma-glutamyl transferase; TP, total protein; ALB, albumin.

**Table 5 nutrients-14-02971-t005:** Lipid profiles in response to the 12-day nutritional intervention with control or Mo-biofortified lettuce. Values are indicated as means ± standard deviations; *p*-values higher than 0.05 mean that the change is not statistically significant, and indicate strong evidence for the null hypothesis.

Parameters	Control Group			Intervention Group		
	Mean ± SD		*p*-Value	Mean ± SD		*p*-Value
	T0	T1		T0	T1	
TG (mg/dL)	84.7± 45.6	81.5 ± 28.6	>0.05	89.0 ± 36.8	87.0 ± 40.9	>0.05
CHOL TOT (mg/dL)	167.9 ± 23.6	166.9 ± 27.8	>0.05	174.2 ± 23.0	173.7 ± 23.0	>0.05
CHOL LDL (mg/dL)	96.9 ± 22.4	97.2 ± 28.0	>0.05	102.1 ± 17.7	104.2 ± 28.8	>0.05
CHOL HDL (mg/dL)	49.1 ± 7.00	47.2 ± 7.5	>0.05	55.3 ± 15.6	56.1 ± 16	>0.05

Acronyms: TG, triglycerides; CHOL, cholesterol; LDL, low-density lipoprotein; HDL, high-density lipoprotein.

**Table 6 nutrients-14-02971-t006:** Serum molybdenum concentrations (μg/L).

Control Group		Intervention Group	
Mean ± SD		Mean ± SD	
T0	T1	T0	T1
4.9 ± 1.6 μg/L	5.1 ± 1.7 μg/L	5.0 ± 1.7 μg/L	7.1* ± 1.5 μg/L

*p*-Values lower than 0.05 mean that the change is statistically significant, and are indicated by asterisks. Asterisks denote significant differences compared to T0 and T1of the control group and T0 of the intervention group with respect to T1 of the intervention group; *n* = 12 in each group.

## Data Availability

The datasets during and/or analyzed during the current study are available from the corresponding author on reasonable request.

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
