# Peer review of "A New Potential Dietary Approach to Supply Micronutrients to Physically Active People through Consumption of Biofortified Vegetables"

_nutrients, 2022, doi:10.3390/nu14142971_

Round 1

Reviewer 1 Report

In this manuscript, the authors reported a new potential dietary approach to verify whether supplementation for twelve days with 25 molybdenum (Mo) fortified vegetables impact the physically active individuals.The research is carried out accurately and the results were successfully obtained. However, there are several issues needed to be addressed.

   Q1: Line 21, check the grammar and spelling agin carefully throughout the manuscript.

   Q2: Abstract needs to be rewritten. The key point and the novelty of the study must be explained. 

   Q3: Line 189, what is the standard solutions? Was the method based on some literature method? If so, those citations should be included here.  

  Q4: The conclusion section was not sufficient to effectively summarize the content of the entire manuscript. This section needs to be more specific and detailed.

Author Response

Thank you very much for your comments and suggestions to improve the quality of our manuscript. We really appreciate the time you dedicated to us.

A1: Line 21 was adjusted to improve readability (now line 50-51). Grammar and spelling were checked throughout the manuscript and adjusted. If we missed something please let us know and we will adjust it immediately.

A2: Abstract was rewritten according to your suggestions to point out key points and the novelty of the study.

A3: The standard solutions are the solutions obtained by diluting the certified reference material. They are used for the construction of the calibration curve. Yes, the method was based on the paper of Cammilleri et al., Nat Prod Res. 2022, 8:1-11. As you required we added the information in line 199.

A4: We have rewritten the conclusion section according to your suggestions to be more specific and detailed.

Reviewer 2 Report

There is a mistake in the table 4, have to correct it, in the  row

CHOL LDL (mg/dL) 96.9 ± 22.4 97.2 ± 28.0 >0.05    02.1 ± 17.7  104.2 ± 28.8 >0.05. 

  • The question original and well-defined,  the results provide an advancement of the current knowledge
  • The work fit the journal scope.
  • The results interpreted appropriatel, They are significant, I prefer  if  mark the significance levels exactly not only in the figure but also in the tables. All conclusions justified and supported by the results.  
  • The article written in an appropriate .  The data and analyses presented appropriately. 
  • The study correctly designed and technically sound. The analyses performed with the highest technical standards. The data robust enough to draw conclusions? The methods, tools, software, and reagents described with sufficient details to allow another researcher to reproduce the results. The raw data available and correct 

Manuscripts submitted to MDPI journals should meet the highest standards of publication ethics:

  • Manuscripts should only report results that have not been submitted or published before, even in part.
  • Manuscripts must be original and should not reuse text from another source without appropriate citation.
  • The studies reported should have been carried out in accordance with generally accepted ethical research standards.

Author Response

Thank you very much for your positive comments to our manuscript.  As you suggested we have adjusted the mistake in table 4. Thanks

Round 2

Reviewer 1 Report

  • Accept in Present Form: The paper can be accepted without any further changes.